# Learning Mahalanobis Metric Spaces via Geometric Approximation Algorithms

## Abstract

Learning Mahalanobis metric spaces is an important problem that has found numerous applications. Several algorithms have been designed for this problem, including Information Theoretic Metric Learning (ITML) [Davis et al. 2007] and Large Margin Nearest Neighbor (LMNN) classification [Weinberger and Saul 2009]. We consider a formulation of Mahalanobis metric learning as an optimization problem, where the objective is to minimize the number of violated similarity/dissimilarity constraints. We show that for any fixed ambient dimension, there exists a fully polynomial-time approximation scheme (FPTAS) with nearly-linear running time. This result is obtained using tools from the theory of linear programming in low dimensions. We also discuss improvements of the algorithm in practice, and present experimental results on synthetic and real-world data sets. Our algorithm is fully parallelizable and performs favorably in the presence of adversarial noise.

## 1 Introduction

Learning metric spaces is a fundamental computational primitive that has found numerous applications and has received significant attention in the literature. We refer the reader to Kulis et al. (2013); Li and Tian (2018) for detailed exposition and discussion of previous work. At the high level, the input to a metric learning problem consists of some universe of objects $X$, together with some similarity information on subsets of these objects. Here, we focus on pairwise similarity and dissimilarity constraints. Specifically, we are given $\mathcal{S}, \mathcal{D} \subset \binom{X}{2}$, which are sets of pairs of objects that are labeled as similar and dissimilar respectively. We are also given some $u, \ell > 0$, and we seek to find a mapping $f : X \to Y$, into some target metric space $(Y, \rho)$, such that for all $x, y \in \mathcal{S}$,

$$\rho(f(x), f(y)) \leqslant u,$$

and for all $x, y \in \mathcal{D}$,

$$\rho(f(x), f(y)) \geqslant \ell.$$

In the case of Mahalanobis metric learning, we have $X \subset \mathbb{R}^d$, with $|X| = n$, for some $d \in \mathbb{N}$, and the mapping $f : \mathbb{R}^d \to \mathbb{R}^d$ is linear. Specifically, we seek to find a matrix $\mathbf{G} \in \mathbb{R}^{d \times d}$, such that for all $\{p, q\} \in \mathcal{S}$, we have

$$\|\mathbf{G}p - \mathbf{G}q\|_2 \leqslant u, \tag{1}$$

and for all $\{p, q\} \in \mathcal{D}$, we have

$$\|\mathbf{G}p - \mathbf{G}q\|_2 \geqslant \ell. \tag{2}$$

### 1.1 Our Contribution

In general, there might not exist any $\mathbf{G}$ that satisfies all constraints of type 1 and 2. We are thus interested in finding a solution that minimizes the fraction of violated constraints, which corresponds to maximizing the accuracy of the mapping. We develop a $(1 + \varepsilon)$-approximation algorithm for optimization problem of computing a Mahalanobis metric space of maximum accuracy, that runs in near-linear time for any fixed ambient dimension $d \in \mathbb{N}$. This algorithm is obtained using tools from geometric approximation algorithms and the theory of linear programming in small dimension. The following summarizes our result.

**Theorem 1.1.** *For any $d \in \mathbb{N}$, $\varepsilon > 0$, there exists a randomized algorithm for learning $d$-dimensional Mahalanobis metric spaces, which given an instance that admits a mapping with accuracy $r^*$, computes a mapping with accuracy at least $r^* - \varepsilon$, in time $d^{O(1)} n (\log n / \varepsilon)^{O(d)}$, with high probability.*

The above algorithm can be extended to handle various forms of regularization. We also propose several modifications of our algorithm that lead to significant performance improvements in practice. The final algorithm is evaluated experimentally on both synthetic and real-world data sets, and is compared against the currently best-known algorithms for the problem.

## 1.2 RELATED WORK

Several algorithms for learning Mahalanobis metric spaces have been proposed. Notable examples include the SDP based algorithm of Xing et al. Xing et al. (2003), the algorithm of Globerson and Roweis for the fully supervised setting Globerson and Roweis (2006), Information Theoretic Metric Learning (ITML) by Davis et al. Davis et al. (2007), which casts the problem as a particular optimization minimizing LogDet divergence, as well as Large Margin Nearest Neighbor (LMNN) by Weinberger et al. Weinberger et al. (2006), which attempts to learn a metric geared towards optimizing $k$-NN classification. We refer the reader to the surveys Kulis et al. (2013); Li and Tian (2018) for a detailed discussion of previous work. Our algorithm differs from previous approaches in that it seeks to directly minimize the number of violated pairwise distance constraints, which is a highly non-convex objective, without resorting to a convex relaxation of the corresponding optimization problem.

## 1.3 ORGANIZATION

The rest of the paper is organized as follows. Section 2 describes the main algorithm and the proof of Theorem 1.1. Section 3 discusses practical improvements used in the implementation of the algorithm. Section 4 presents the experimental evaluation.

## 2 MAHALANOBIS METRIC LEARNING AS AN LP-TYPE PROBLEM

In this Section we present an approximation scheme for Mahalanobis metric learning in $d$-dimensional Euclidean space, with nearly-linear running time. We begin by recalling some prior results on the class of LP-type problems, which generalizes linear programming. We then show that linear metric learning can be cast as an LP-type problem.

### 2.1 LP-TYPE PROBLEMS

Let us recall the definition of an LP-type problem. Let $\mathcal{H}$ be a set of constraints, and let $w : 2^{\mathcal{H}} \to \mathbb{R} \cup \{-\infty, +\infty\}$, such that for any $G \subset \mathcal{H}$, $w(G)$ is the value of the optimal solution of the instance defined by $G$. We say that $(\mathcal{H}, w)$ defines an LP-type problem if the following axioms hold:

**(A1) Monotonicity.** For any $F \subseteq G \subseteq \mathcal{H}$, we have $w(F) \leqslant w(G)$.

**(A2) Locality.** For any $F \subseteq G \subseteq \mathcal{H}$, with $-\infty < w(F) = w(G)$, and any $h \in \mathcal{H}$, if $w(G) < w(G \cup \{h\})$, then $w(F) < w(F \cup \{h\})$.

More generally, we say that $(\mathcal{H}, w)$ defines an LP-type problem on some $\mathcal{H}' \subseteq \mathcal{H}$, when conditions (A1) and (A2) hold for all $F \subseteq G \subseteq \mathcal{H}'$.

A subset $B \subseteq \mathcal{H}$ is called a *basis* if $w(B) > -\infty$ and $w(B') < w(B)$ for any proper subset $B' \subsetneq B$. A *basic operation* is defined to be one of the following:

**(B0) Initial basis computation.** Given some $G \subseteq \mathcal{H}$, compute any basis for $\mathcal{G}$.

**(B1) Violation test.** For some $h \in \mathcal{H}$ and some basis $B \subseteq \mathcal{H}$, test whether $w(B \cup \{h\}) > w(B)$ (in other words, whether $B$ violates $h$).

**(B2) Basis computation.** For some $h \in \mathcal{H}$ and some basis $B \subseteq \mathcal{H}$, compute a basis of $B \cup \{h\}$.

## 2.2 An LP-type Formulation

We now show that learning Mahalanobis metric spaces can be expressed as an LP-type problem. We first note that we can rewrite (1) and (2) as

$$(p-q)^T \mathbf{A}(p-q) \leqslant u^2, \tag{3}$$

and

$$(p-q)^T \mathbf{A}(p-q) \geqslant \ell^2, \tag{4}$$

where $\mathbf{A} = \mathbf{G}^T \mathbf{G}$ is positive semidefinite.

We define $\mathcal{H} = \{0, 1\} \times \binom{\mathbb{R}^d}{2}$, where for each $(0, \{p, q\}) \in \mathcal{H}$, we have a constraint of type (3), and for every $(1, \{p, q\}) \in \mathcal{H}$, we have a constraint of type (4). Therefore, for any set of constraints $F \subseteq \mathcal{H}$, we may associate the set of feasible solutions for $F$ with the set $\mathcal{A}_F$ of all positive semidefinite matrices $\mathbf{A} \in \mathbb{R}^{n \times n}$, satisfying (3) and (4) for all constraints in $F$.

Let $w : 2^{\mathcal{H}} \to \mathbb{R}$, such that for all $F \in \mathcal{H}$, we have

$$w(F) = \begin{cases} \inf_{\mathbf{A} \in \mathcal{A}_F} r^T \mathbf{A} r & \text{if } \mathcal{A}_F \neq \varnothing \\ \infty & \text{if } \mathcal{A}_F = \varnothing \end{cases},$$

where $r \in \mathbb{R}^d$ is a vector chosen uniformly at random from the unit sphere from some rotationally-invariant probability measure. Such a vector can be chosen, for example, by first choosing some $r' \in \mathbb{R}^d$, where each coordinate is sampled from the normal distribution $\mathcal{N}(0, 1)$, and setting $r = r'/\|r'\|_2$.

**Lemma 2.1.** *When $w$ is chosen as above, the pair $(\mathcal{H}, w)$ defines an LP-type problem of combinatorial dimension $O(d^2)$, with probability 1. Moreover, for any $n > 0$, if each $r_i$ is chosen using $\Omega(\log n)$ bits of precision, then for each $F \subseteq \mathcal{H}$, with $n = |F|$, the assertion holds with high probability.*

*Proof.* Since adding constraints to a feasible instance can only make it infeasible, it follows that $w$ satisfies the monotonicity axiom (A1).

We next argue that the locality axion (A2) also holds, with high probability. Let $F \subseteq G \subseteq \mathcal{H}$, with $-\infty < w(F) = w(G)$, and let $h \in \mathcal{H}$, with $w(G) < w(G \cup \{h\})$. Let $\mathbf{A}_F \in \mathcal{A}_F$ and $\mathbf{A}_G \in \mathcal{A}_G$ be some (not necessarily unique) infimizers of $w(\mathbf{A})$, when $\mathbf{A}$ ranges in $\mathcal{A}_F$ and $\mathcal{A}_G$ respectively. The set $\mathcal{A}_F$, viewed as a convex subset of $\mathbb{R}^{d^2}$, is the intersection of the SDP cone with $n$ half-spaces, and thus $\mathcal{A}_F$ has at most $n$ facets. There are at least two distinct infimizers for $w(\mathbf{A}_G)$, when $\mathbf{A}_G \in \mathcal{A}_G$, only when the randomly chosen vector $r$ is orthogonal to a certain direction, which occurs with probability 0. When each entry of $r$ is chosen with $c \log n$ bits of precision, the probability that $r$ is orthogonal to any single hyperplane is at most $2^{-c \log n} = n^{-c}$; the assertion follows by a union bound over $n$ facets. This establishes that axiom (A2) holds with high probability.

It remains to bound the combinatorial dimension, $\kappa$. Let $F \subseteq \mathcal{H}$ be a set of constraints. For each $\mathbf{A} \in \mathcal{A}_F$, define the ellipsoid

$$\mathcal{E}_{\mathbf{A}} = \{v \in \mathbb{R}^d : \|\mathbf{A}v\|_2 = 1\}.$$

For any $\mathbf{A}, \mathbf{A}' \in \mathcal{A}_F$, with $\mathcal{E}_{\mathbf{A}} = \mathcal{E}_{\mathbf{A}'}$, and $\mathbf{A} = \mathbf{G}^T \mathbf{G}$, $\mathbf{A}' = \mathbf{G}'^T \mathbf{G}'$, we have that for all $p, q \in \mathbb{R}^d$, $\|\mathbf{G}p - \mathbf{G}q\|_2 = (p-q)^T \mathbf{A}(p-q) = (p-q)^T \mathbf{A}'(p-q) = \|\mathbf{G}'p - \mathbf{G}'q\|_2$. Therefore in order to specify a linear transformation $\mathbf{G}$, up to an isometry, it suffices to specify the ellipsoid $\mathcal{E}_{\mathbf{A}}$.

Each $\{p, q\} \in \mathcal{S}$ corresponds to the constraint that the point $(p-q)/u$ must lie in $\mathcal{E}_{\mathbf{A}}$. Similarly each $\{p, q\} \in \mathcal{D}$ corresponds to the constraint that the point $(p-q)/\ell$ must lie either on the boundary or the exterior of $\mathcal{E}_{\mathbf{A}}$. Any ellipsoid in $\mathbb{R}^d$ is uniquely determined by specifying at most $(d+3)d/2 = O(d^2)$ distinct points on its boundary (see Welzl (1991); Chazelle (2000)). Therefore, each optimal solution can be uniquely specified as the intersection of at most $O(d^2)$ constraints, and thus the combinatorial dimension is $O(d^2)$. $\qquad\square$

**Lemma 2.2.** *Any initial basis computation (B0), any violation test (B1), and any basis computation (B2) can be performed in time $d^{O(1)}$.*

*Proof.* The violation test (B1) can be performed by solving one SDP to compute $w(B)$, and another to compute $w(B \cup \{h\})$. By Lemma 2.1 the combinatorial dimension is $O(d^2)$, thus each SDP has $O(d^2)$ constraints, and be solved in time $d^{O(1)}$.

The basis computation step (B2) can be performed starting with the set of constraints $B \cup \{h\}$, and iteratively remove every constraint whose removal does not decrease the optimum cost, until we arrive at a minimal set, which is a basis. In total, we need to solve at most $d$ SDPs, each of size $O(d^2)$, which can be done in total time $d^{O(1)}$.

Finally, by the choice of $w$, any set containing a single constraint in $\mathcal{S}$ is a valid initial basis. $\qquad\square$

### 2.3 ALGORITHMIC IMPLICATIONS

Using the above formulation of Mahalanobis metric learning as an LP-type problem, we can obtain our approximation scheme. Our algorithm uses as a subroutine an *exact* algorithm for the problem (that is, for the special case where we seek to find a mapping that satisfies all constraints). We first present the exact algorithm and then show how it can be used to derive the approximation scheme.

**An exact algorithm.** Welzl (1991) obtained a simple randomized linear-time algorithm for the minimum enclosing ball and minimum enclosing ellipsoid problems. This algorithm naturally extends to general LP-type problems (we refer the reader to Har-Peled (2011); Chazelle (2000) for further details).

With the interpretation of Mahalanobis metric learning as an LP-type problem given above, we thus obtain a linear time algorithm for in $\mathbb{R}^d$, for any constant $d \in \mathbb{N}$. The resulting algorithm on a set of constraints $F \subseteq \mathcal{H}$ is implemented by the procedure Exact-LPTML$(F; \varnothing)$, which is presented in Algorithm 1. The procedure LPTML$(F; B)$ takes as input sets of constraints $F, B \subseteq \mathcal{H}$. It outputs a solution $\mathbf{A} \in \mathbb{R}^{d \times d}$ to the problem induced by the set of constraints $F \cup B$, such that all constraints in $B$ are tight (that is, they hold with equality); if no such solution solution exists, then it returns nil. The procedure Basic-LPTML$(B)$ computes LPTML$(\varnothing; B)$. The analysis of Welzl (1991) implies that when Basic-LPTML$(B)$ is called, the cardinality of $B$ is at most the combinatorial dimension, which by Lemma 2.1 is $O(d^2)$. Thus the procedure Basic-LPTML can be implemented using one initial basis computation (B0) and $O(d^2)$ basis computations (B2), which by Lemma 2.2 takes total time $d^{O(1)}$.

---

**Algorithm 1** An exact algorithm for Mahalanobis metric learning.

---

    **procedure** Exact-LPTML$(F; B)$
        **if** $F = \varnothing$ **then**
            $\mathbf{A} \leftarrow$ Basic-LPTML$(B)$
        **else**
            choose $h \in F$ uniformly at random
            $\mathbf{A} \leftarrow$ Exact-LPTML$(F - \{h\})$
            **if** $\mathbf{A}$ violates $h$ **then**
                $\mathbf{A} :=$ Exact-LPTML$(F - \{h\}; B \cup \{h\})$
            **end if**
        **end if**
        **return A**
    **end procedure**

---

**An $(1 + \varepsilon)$-approximation algorithm.** It is known that the above exact linear-time algorithm leads to an nearly-linear-time approximation scheme for LP-type problems. This is summarized in the following. We refer the reader to Har-Peled (2011) for a more detailed treatment.

**Lemma 2.3** (Har-Peled (2011), Ch. 15). *Let $\mathcal{A}$ be some LP-type problem of combinatorial dimension $\kappa > 0$, defined by some pair $(\mathcal{H}, w)$, and let $\varepsilon > 0$. There exists a randomized algorithm which given some instance $F \subseteq \mathcal{H}$, with $|F| = n$, outputs some basis $B \subseteq F$, that violates at most $(1 + \varepsilon)k$ constraints in $F$, such that $w(B) \leqslant w(B')$, for any basis $B'$ violating at most $k$ constraints in $F$, in time $O\left(t_0 + \left(n + n \min\left\{\frac{\log^{\kappa+1} n}{\varepsilon^{2\kappa}}, \frac{\log^{\kappa+2} n}{k\varepsilon^{2\kappa+2}}\right\}\right)(t_1 + t_2)\right)$, where $t_0$ is the time needed to compute*

*an arbitrary initial basis of $\mathcal{A}$, and $t_1$, $t_2$, and $t_3$ are upper bounds on the time needed to perform the basic operations (B0), (B1) and (B2) respectively. The algorithm succeeds with high probability.*

For the special case of Mahalanobis metric learning, the corresponding algorithm is given in Algorithm 2. The approximation guarantee for this algorithm is summarized in 1.1. We can now give the proof of our main result.

*Proof of Theorem 1.1.* Follows immediately by Lemmas 2.2 and 2.3. □

---

**Algorithm 2** An approximation algorithm for Mahalanobis metric learning.

---

**procedure** LPTML($F$)
    **for** $i = 0$ to $\log_{1+\varepsilon} n$ **do**
        $p \leftarrow (1 + \varepsilon)^{-i}$
        **for** $j = 1$ to $\log^{O(d^2)} n$ **do**
            subsample $F_j \subseteq F$, where each element is chosen independently with probability $p$
            $\mathbf{A}_{i,j} \leftarrow$ Exact-LPTML($F_j$)
        **end for**
    **end for**
    **return** a solution out of $\{\mathbf{A}_{i,j}\}_{i,j}$, violating the minimum number of constraints in $F$
**end procedure**

---

**Regularization.** We now argue that the LP-type algorithm described above can be extended to handle certain types of regularization on the matrix $\mathbf{A}$. In methods based on convex optimization, introducing regularizers that are convex functions can often be done easily. In our case, we cannot directly introduce a regularizing term in the objective function that is implicit in Algorithm 2. More specifically, let $\text{cost}(\mathbf{A})$ denote the total number of constraints of type (3) and (4) that $\mathbf{A}$ violates. Algorithm 2 approximately minimizes the objective function $\text{cost}(\mathbf{A})$. A natural regularized version of Mahalanobis metric learning is to instead minimize the objective function

$$\text{cost}'(\mathbf{A}) := \text{cost}(\mathbf{A}) + \eta \cdot \text{reg}(\mathbf{A}),$$

for some $\eta > 0$, and regularizer $\text{reg}(\mathbf{A})$. One typical choice is $\text{reg}(\mathbf{A}) = \text{tr}(\mathbf{AC})$, for some matrix $\mathbf{C} \in \mathbb{R}^{d \times d}$; the case $\mathbf{C} = \mathbf{I}$ corresponds to the trace norm (see Kulis et al. (2013)). We can extend the Algorithm 2 to handle any regularizer that can be expressed as a linear function on the entries of $\mathbf{A}$, such as $\text{tr}(\mathbf{A})$. The following summarizes the result.

**Theorem 2.4.** *Let $\text{reg}(\mathbf{A})$ be a linear function on the entries of $\mathbf{A}$, with polynomially bounded coefficients. For any $d \in \mathbb{N}$, $\varepsilon > 0$, there exists a randomized algorithm for learning $d$-dimensional Mahalanobis metric spaces, which given an instance that admits a solution $\mathbf{A}_0$ with $\text{cost}'(\mathbf{A}_0) = c^*$, computes a solution $\mathbf{A}$ with $\text{cost}'(\mathbf{A}) \leqslant (1+\varepsilon)c^*$, in time $d^{O(1)} n (\log n / \varepsilon)^{O(d)}$, with high probability.*

*Proof.* If $\eta < \varepsilon^t$, for sufficiently large constant $t > 0$, since the coefficients in $\text{reg}(\mathbf{A})$ are polynomially bounded, it follows that the largest possible value of $\eta \cdot \text{reg}(\mathbf{A})$ is $O(\varepsilon)$, and can thus be omitted without affecting the result. Similarly, if $\eta > (1/\varepsilon)n^{t'}$, for sufficiently large constant $t' > 0$, since there are at most $\binom{n}{2}$ constraints, it follows that the term $\text{cost}(\mathbf{A})$ can be omitted form the objective. Therefore, we may assume w.l.o.g. that $\text{reg}(A_0) \in [\varepsilon^{O(1)}, (1/\varepsilon)n^{O(1)}]$. We can guess some $i = O(\log n + \log(1/\varepsilon))$, such that $\text{reg}(A_0) \in ((1+\varepsilon)^{i-1}, (1+\varepsilon)^i]$. We modify the SDP used in the proof of Lemma 2.2 by introducing the constraint $\text{reg}(\mathbf{A}) \leqslant (1 + \varepsilon)^i$. Guessing the correct value of $i$ requires $O(\log n + \log(1/\varepsilon))$ executions of Algorithm 2, which implies the running time bound. □

## 3   Practical Improvements and Parallelization

We now discuss some modifications of the algorithm described in the previous section that significantly improve its performance in practical scenarios, and have been integrated in our implementation.

**Move-to-front and pivoting heuristics.** We use heuristics that have been previously used in algorithms for linear programming Seidel (1990); Clarkson (1995), minimum enclosing ball in $\mathbb{R}^3$ Megiddo (1983), minimum enclosing ball and ellipsoid is $\mathbb{R}^d$, for any fixed $d \in \mathbb{N}$ Welzl (1991), as well as in fast implementations of minimum enclosing ball algorithms Gärtner (1999). The *move-to-front* heuristic keeps an ordered list of constraints which gets reorganized as the algorithm runs; when the algorithm finds a violation, it moves the violating constraint to the beginning of the list of the current sub-problem. The *pivoting* heuristic further improves performance by choosing to add to the basis the constraint that is "violated the most". For instance, for similarity constraints, we pick the one that is mapped to the largest distance greater than $u$; for dissimilarity constraints, we pick the one that is mapped to the smallest distance less than $\ell$.

**Approximate counting.** The main loop of Algorithm 2 involves counting the number of violated constraints in each iteration. In problems involving a large number of constraints, we use approximate counting by only counting the number of violations within a sample of $O(\log 1/\varepsilon)$ constraints. We denote by LPTML$_t$ for the version of the algorithm that performs a total of $t$ iterations of the inner loop.

**Early termination.** A bottleneck of Algorithm 2 stems from the fact that the inner loop needs to be executed for $\log^{O(d^2)} n$ iterations. In practice, we have observed that a significantly smaller number of iterations is needed to achieve high accuracy.

**Parallelization.** Algorithm 2 consists of several executions of the algorithm Exact-LPTML on independently sampled sub-problems. Therefore, Algorithm 2 can trivially be parallelized by distributing a different set of sub-problems to each machine, and returning the best solution found overall.

## 4 EXPERIMENTAL EVALUATION

We have implemented Algorithm 2, incorporating the practical improvements described in Section 3, and performed experiments on synthetic and real-world data sets. Our LPTML implementation and documentation can be found at the supplementary material[1]. We now describe the experimental setting and discuss the main findings.

### 4.1 EXPERIMENTAL SETTING

**Classification task.** Each data set used in the experiments consists of a set of labeled points in $\mathbb{R}^d$. The label of each point indicates its class, and there is a constant number of classes. The set of similarity constraints $\mathcal{S}$ (respt. dissimilarity constraints $\mathcal{D}$) is formed by uniformly sampling pairs of points in the same class (resp. from different classes). We use various algorithms to learn a Mahalanobis metric for a labeled input point set in $\mathbb{R}^d$, given these constraints. The values $u$ and $\ell$ are chosen as the 90th and 10th percentiles of all pairwise distances. We used 2-fold cross-validation: At the training phase we learn a Mahalanobis metric, and in the testing phase we use $k$-NN classification, with $k = 4$, to evaluate the performance.

**Data sets.** We have tested our algorithm on the following synthetic and real-world data sets:

1. *Real-world:* We have tested the performance of our implementation on the Iris, Wine, Ionosphere and Soybean data sets from the UCI Machine Learning Repository[2].

2. *Synthetic:* Next, we consider a synthetic data set that is constructed by first sampling a set of 100 points from a mixture of two Gaussians in $\mathbb{R}^2$, with identity covariance matrices, and with means $(-3, 0)$ and $(3, 0)$ respectively; we then apply a linear transformation that stretches the $y$ axis by a factor of $40$. This linear transformation reduces the accuracy of $k$-NN on the underlying Euclidean metric with $k = 4$ from 1 to 0.68.

---

[1] A copy of our implementation is also available at `https://drive.google.com/drive/folders/1XgABqyh8E1CoRGadh1KC5or7TBLgQkdl?usp=sharing`

[2] `https://archive.ics.uci.edu/ml/datasets.php`

*3. Synthetic + Adversarial Noise:* We modify the above synthetic data set by introducing a small fraction of points in an adversarial manner, before applying the linear transformation. Figure 3b depicts the noise added as five points labeled as one of the classes, and sampled from a Gaussian with identity covariance matrix and mean $(-100, 0)$ (Figure 3a).

**Algorithms.**   We compare the performance of our algorithm against ITML and LMNN. We used the implementations provided by the authors of these works, with minor modifications.

## 4.2   RESULTS

**Accuracy.**   Algorithm 2 minimizes the number of violated pairwise distance constraints. It is interesting to examine the effect of this objective function on the accuracy of $k$-NN classification. Figure 1 depicts this relationship for the Wine data set. We observe that, in general, as the number of iterations of the main loop of LPTML increases, the number of violated pairwise distance constraints decreases, and the accuracy of $k$-NN increases. This phenomenon remains consistent when we first perform PCA to $d = 4, 8, 12$ dimensions.

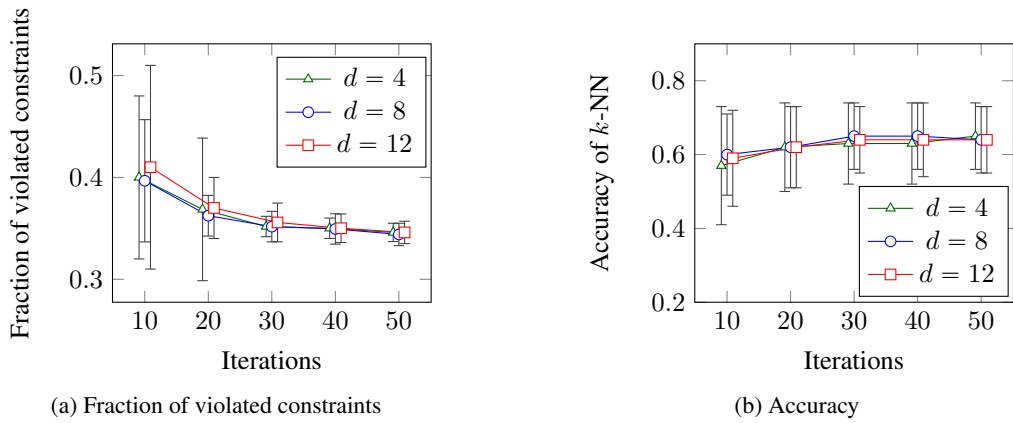

(a) Fraction of violated constraints                    (b) Accuracy

Figure 1: Fraction of violated constraints (left) and accuracy (right) as the number of iterations of LPTML increases (average over 10 executions). Each curve corresponds to the Wine data reduced to $d$ dimensions using PCA.

**Comparison to ITML and LMNN.**   We compared the accuracy obtained by $\text{LPTML}_t$, for $t = 2000$ iterations, against ITML and LMNN. Table 1 summarizes the findings on the real-world and data sets and the synthetic data set without adversarial noise. We observe that LPTML achieves accuracy that is comparable to ITML and LMNN.

We observe that LPTML outperforms ITML and LMNN on the Synthetic + Adversarial Noise data set. This is due to the fact that the introduction of adversarial noise causes the relaxations used in ITML and LMNN to be biased towards contracting the $x$-axis. In contrast, the noise does not "fool" LPTML because it only changes the optimal accuracy by a small amount. The results are summarized in Figure 2.

| Data set | ITML | LMNN | $\text{LPTML}_{t=2000}$ |
|---|---|---|---|
| Iris | $0.96 \pm 0.01$ | $0.96 \pm 0.02$ | $0.94 \pm 0.04$ |
| Soybean | $0.95 \pm 0.04$ | $0.96 \pm 0.04$ | $0.90 \pm 0.05$ |
| Synthetic | $0.97 \pm 0.02$ | $1.00 \pm 0.00$ | $1.00 \pm 0.00$ |

Table 1: Average accuracy and standard deviation over 50 executions of ITML, LMNN and LPTML.

**The effect of dimension.**   The running time of LPTML grows with the dimension $d$. This is caused mostly by the fact that the combinatorial dimension of the underlying LP-type problem is $O(d^2)$, and thus performing each basic operation requires solving an SDP with $O(d^2)$ constraints.

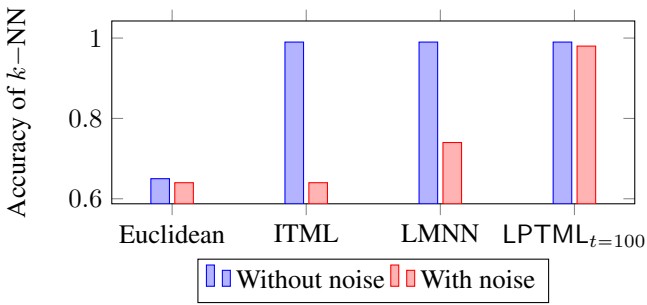

Figure 2: Accuracy in a syntetic data set scenario where a small fraction of points were placed adversarially. See Figure 3a.

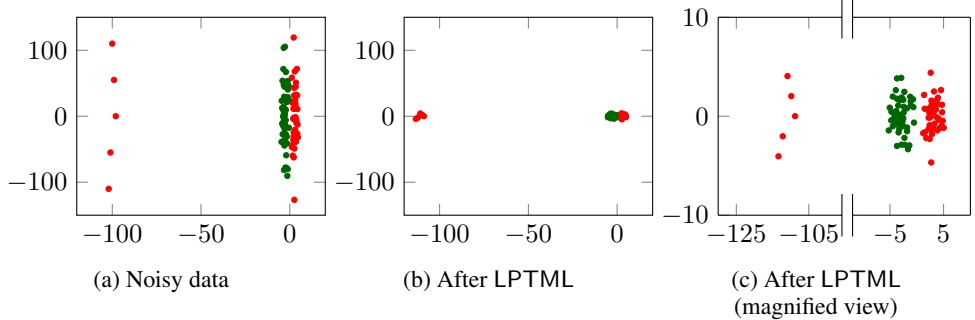

(a) Noisy data      (b) After LPTML      (c) After LPTML (magnified view)

Figure 3: Synthetic + Adversarial Noise scenario. On the left, the data set before learning. The center image shows the correct transformation recovered by LPTML. On the right, a magnified view of the recovered data.

Figure 4 (appearing in the Appendix) depicts the effect of dimensionality in the running time, for $t = 100, \ldots, 2000$ iterations of the main loop. The data set used is Wine after performing PCA to $d$ dimensions, for $d = 2, \ldots, 13$.

**Parallel implementation.** We implemented a massively parallel version of LPTML in the MapReduce model. The program maps different sub-problems of the main loop of LPTML to different machines. In the reduce step, we keep the result with the minimum number of constraint violations. The implementation uses the mrjob Yelp and Contributors (2019) package. For these experiments, we used Amazon cloud computing instances of type *m4.xlarge*, AMI 5.20.0 and configured with Hadoop. As expected, the training time decreases as the number of available processors increases (Figure 5 in the Appendix). All technical details about this implementation can be found in the parallel section of the documentation of our code.

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

## A ADDITIONAL FIGURES

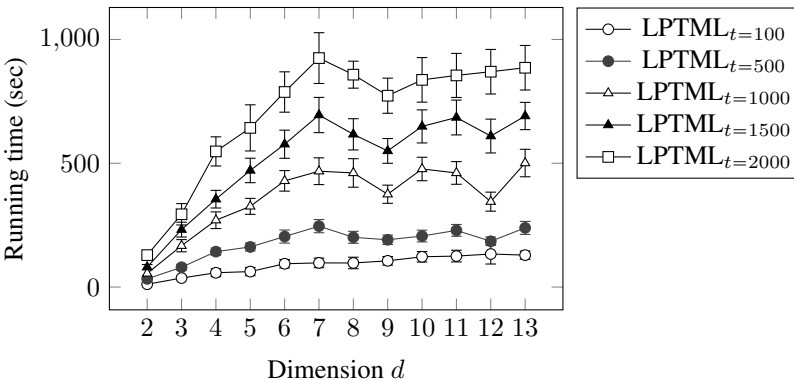

Figure 4: Average running time of 10 executions of LPTML at different levels of dimensionality reduction using PCA on the Wine data set. Each curve corresponds to LPTML limited to $t$ iterations.

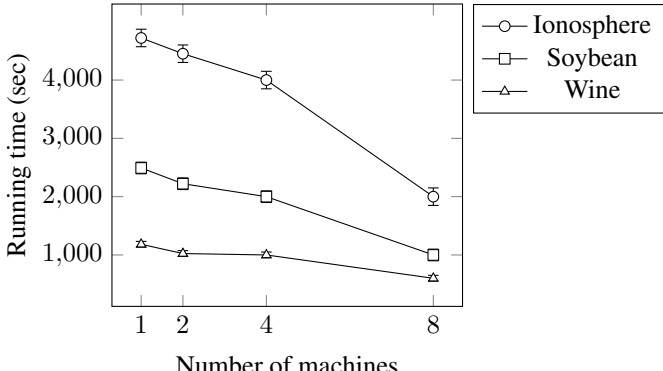

Figure 5: Running time for parallel LPTML on an increasing number of machines (average over 10 executions).

