# OpenReview forum: "Learning Mahalanobis Metric Spaces via Geometric Approximation Algorithms"
_ICLR.cc/2020/Conference — Reject_

### Official Review · AnonReviewer1 · 2019-10-21
**Official Blind Review #1**

**Rating:** 3

**Review:**

The paper proposes a method to handle Mahalanobis metric learning thorough linear programming. The authors consider the specific setup where examples are labelled as similar or dissimilar and the task is to find a mapping such that the feature-space distance between examples is i) smaller than a certain value if the examples are labelled as 'similar' and ii) greater than a possibly different value if the examples are labelled as 'dissimilar'. Arguments from the theory of linear programming are leveraged to define exact and approximated algorithms.

I would tend to reject the paper because I do not fully understand where is where is the main novelty.  Transforming the problem into a linear programming does not look a very complicated step given the specific setup considered in the paper. Moreover, it is not clear enough if there are computational or theoretical gains in following the proposed approach instead of applying other existing methods. Especially because the provided experiments seem to show that there is no improvement  in the accuracy, the authors should have spent some more words  to motivate their strategy.

Questions:

- Are there any theoretical guarantees for the proposed approximation? Is the proposed approximation strategy completely new or similar approaches have been already applied to slightly different setups?

- What are the key differences between the proposed method and with other convex approximations for learning Mahalanobis metrics? As the experimental performance of the proposed approach and other existing methods, what are the net advantages to be associated with the geometric approximation?

- Why is the approximation needed?

- In the proof, why is it true that, for a given solution, adding a constraint implies this constraint is not satisfied?



**Experience Assessment:**

I have read many papers in this area.

**Review Assessment: Checking Correctness Of Derivations And Theory:**

I assessed the sensibility of the derivations and theory.

**Review Assessment: Checking Correctness Of Experiments:**

I assessed the sensibility of the experiments.

**Review Assessment: Thoroughness In Paper Reading:**

I read the paper at least twice and used my best judgement in assessing the paper.

---

> ### Author Response · Authors · 2019-11-15
> **Response**
>
> We thank the reviewer for the insightful comments. Below are our responses to the specific points raised:
>
> - Our algorithm is the first with a provable guarantee on the number of violated constraints on arbitrary (that is, adversarial) inputs. The machinery for solving LP-type problems is well-known within the computational geometry community, but it had never been applied in the context of metric learning prior to our work.
>
> - Prior works are based on minimizing an error function that penalizes violations, which is different than minimizing the number of violations.  The benefit of minimizing the number of violations directly is demonstrated in Figure 3. There, it is shown that a simple adversarial input can fool the previous state-of-the-art on the problem. In contrast, our algorithm correctly learns the ground truth.
>
> - To the best of our knowledge, the exact complexity of the problem is not known. We suspect that the problem is NP-hard when the dimension d is unbounded.
>
> - The phrasing in the proof is confusing. What we mean is that adding a constraint to a feasible instance can make it either feasible or infeasible; adding a constrant to an infeasible instance cannot change its feasibility. We will rephrase this in the final version of our paper.
> We thank the reviewer for the insightful comments. Below are our responses to the specific points raised:
>
> - Our algorithm is the first with a provable guarantee on the number of violated constraints on arbitrary (that is, adversarial) inputs. The machinery for solving LP-type problems is well-known within the computational geometry comminity, but it had never been applied in the context of metric learning prior to our work.
>
> - Pior works are based on minimizing an error function that penalizes violations, which is different than minimizing the number ofs violations.  The benefit of minimizing the number of violations directly is demonstrated in Figure 3. There, it is shown that a simple adversarial input can fool the previous state-of-the-art on the problem. In contrast, our algorithm correctly learns the ground truth.
>
> - To the best of our knowledge, the exact complexity of the problem is not known. We suspect that the problem is NP-hard when the dimension d is unbounded.
>
> - The phrasing in the proof is confusing. What we mean is that adding a constraint to a feasible instance can make it either feasible or infeasible; adding a constraint to an infeasible instance cannot change its feasibility. We will rephrase this in the final version of our paper.

---

### Official Review · AnonReviewer2 · 2019-10-22
**Official Blind Review #2**

**Rating:** 6

**Review:**

This paper discusses the following problem. Given a set X \subset R^d of points, sets S, D of pairs (S and D denoting similar and dissimilar pairs), numbers u, l, find a matrix G such that
- for all pairs (p, q) \in S: ||Gp - Gq|| \leq u, and
- for all pairs (p, q) \in D: ||Gp - Gq|| \geq l.

Note that such a matrix G may not exist for a given input instance (X, S, D, u, l). So, the relevant problem is maximising the number of constraints. The paper gives an (1+\eps)-approximation algorithm for the maximisation problem. The main idea is defining an LP-type problem and then using a previous result of Har-peled. Some experimental results for a heuristic version are given and compared against other Mahalanobis distance learning algorithms (that may or may not be defined as a maximisation problem). I think the paper ports an interesting result from LP-type problems into the context of distance learning that people may find interesting and may encourage further work.

Other comments:
1. I did not find any comment about the computational hardness of the problem. It is always good to the hardness of a problem before evaluating an approximation algorithm for the problem.
2. It will be good to define “accuracy” before using it in Theorem 1.1.
3. Did you define combinatorial dimension before using this in Lemma 2.1?
4. What is Exact-LPTML(.) in line 6 of Algorithm 1. The function call should take two inputs.
5. When you say that your algorithm is an FPTAS I think you are assuming that the dimension d is a constant. It will be good to make this clear.
6. It will good to know what results for minimising the number of constraints are known from past work. The paper mentions some references. It will be much easier if the results that are known about this problem is clearly stated.

**Experience Assessment:**

I do not know much about this area.

**Review Assessment: Checking Correctness Of Derivations And Theory:**

I assessed the sensibility of the derivations and theory.

**Review Assessment: Checking Correctness Of Experiments:**

I assessed the sensibility of the experiments.

**Review Assessment: Thoroughness In Paper Reading:**

I read the paper at least twice and used my best judgement in assessing the paper.

---

> ### Author Response · Authors · 2019-11-15
> **Response**
>
> We thank the reviewer for the insightful comments. Below are our responses to the specific points raised:
>
> 1. To the best of our knowledge, the exact complexity of the problem is not known. We suspect that the problem is NP-hard when the dimension d is unbounded. For constant d, our methods can be used to obtain an exact (that is, optimal) algorithm with running time n^O(d^2). However, since the running time is prohibitively large even for small d, such a result is mostly of theoretical interest.
>
> 2. This is a good suggestion. We will change the final version of the paper appropriately.
>
> 3. This is an omission. The combinatorial dimension is defined to be the maximum cardinality of any basis. We will include this definition to the final version of the paper.
>
> 4. This is a minor typographical error. The second input should be the empty set. The first line of procedure Exact-LPTML also contains a typographical error. It should be "if B = \emptyset", instead of "if F = \emptyset". We will update the final version of the paper accordingly.
>
> 5. We will add this clarification to the final version of the paper.
>
> 6. Our algorithm is the first with a provable guarantee on the number of violated constraints.

---

### Official Review · AnonReviewer3 · 2019-10-25
**Official Blind Review #3**

**Rating:** 3

**Review:**

This paper study the problem of Mahalanobis Metric Spaces learning, which is formulated as an optimization problem with the objective of minimizing the number of violated similarity/dissimilarity constraints.

I am not an expert in this subarea. From what I have read, the method is based on sound theory and outperforms some classical methods, including ITML and LMNN on several standard data sets. However it is unclear to me what is the state-of-the-art of this field from this paper and its novelty.

Some recent papers might worth discussing and comparing with, e.g.,
Verma and Branson, Sample Complexity of Learning Mahalanobis Distance Metrics, NIPS2015
Ye et al. Learning Mahalanobis Distance Metric: Considering Instance Disturbance Helps. IJCAI

In the proof for Lemma2.1., why “adding constraints to a feasible instance can only make it infeasible”?

In Figure 4, why the running time is not a monotonic curve as the dimension increases?

The conclusion of the paper is missing.


**Experience Assessment:**

I have read many papers in this area.

**Review Assessment: Checking Correctness Of Derivations And Theory:**

I assessed the sensibility of the derivations and theory.

**Review Assessment: Checking Correctness Of Experiments:**

I assessed the sensibility of the experiments.

**Review Assessment: Thoroughness In Paper Reading:**

I made a quick assessment of this paper.

---

> ### Author Response · Authors · 2019-11-15
> **Response**
>
> We thank the reviewer for the insightful comments. Below are our responses to the specific points raised:
>
> Regarding the state-of-the-art: ITML and LMNN are the most widely used and cited methods for learning Mahalanobis metrics, and they represent the state-of-the-art for this problem.
>
> Regarding the novelty of our contribution. Our paper gives the first polynomial-time algorithm with a provable near-optimal guarantee on the number of violated constraints, and for arbitrary (that is, adversarial) inputs. We remark that minimizing the number of violated constraints is a highly non-convex constraint. Therefore, methods that are based on convex optimization are not directly applicable in our setting.
>
> The paper of Verma and Branson obtains bounds on the sample complexity of learning a Mahalanobis metric, when the input consists of a set of randomly (i.i.d.) chosen labeled pairs of points from an unknown distribution of bounded support. This is different from our setting where the input is adversarial, and the goal is to obtain an algorithm with provably near-optimal error, and provably fast running time. It is also important to note that the notion of error considered by Verma and Branson is different.
>
> The paper of Ye et al. considers the problem of learning a Mahalanobis metric under perturbations of the input. This setting is completely different than the one considered in our paper.
>
> Regarding the proof of Lemma 2.1. The phrasing in the proof is confusing. What we mean is that adding a constraint to a feasible instance can make it either feasible or infeasible; adding a constraint to an infeasible instance cannot change its feasibility. We will rephrase this in the final version of our paper.
>
> Regarding Figure 4: The running time is not a monotonically increasing function of the dimension because PCA can change the combinatorial structure of an instance in ways that are hard to predict. For example, the minimum number of violations does not need to be a monotonic function of the dimension.

---

### Decision · Program_Chairs · 2019-12-19

**Decision:**

Reject

**Comment:**

The paper proposes a method to handle Mahalanobis metric learning thorough linear programming.

All reviewers were unclear on what novelty of the approach is compared to existing work.

I recommend rejection at this time, but encourage the authors to incorporate reviewers' feedback (in particular placing the work in better context and clarifying the motivations) and resubmitting elsewhere.